# TimeRAG: It's Time for Retrieval-Augmented Generation in Time-Series Forecasting

## Abstract

Time-series data are essential for forecasting tasks across various domains. While Large Language Models (LLMs) have excelled in many areas, they encounter significant challenges in time-series forecasting, particularly in extracting relevant information from extensive temporal datasets. Unlike textual data, time-series data lack explicit retrieval ground truths, complicating the retrieval process. To tackle these issues, we present TimeRAG, a novel retrieval-augmented approach tailored for time-series forecasting. Our method uniquely applies to continuous and complex temporal sequences, and it is trained using LLM feedback, effectively addressing the absence of ground truth and aligning the priorities of the retriever and the LLM. Experimental results demonstrate the effectiveness of TimeRAG, highlighting its ability to significantly enhance forecasting performance and showcasing the potential of LLMs in time-series prediction tasks.

## 1 Introduction

Time-series data are fundamental for forecasting tasks across a broad range of domains, including weather prediction, energy consumption, healthcare monitoring, and financial markets (Yuan et al., 2024). For instance, meteorologists rely on historical climate data to forecast future weather conditions (Govett et al., 2024), energy providers predict demand based on past consumption patterns (Afzal et al., 2024), and healthcare professionals monitor patient vital signs over time to anticipate disease progression (Reed et al., 2005). In financial markets, time-series data such as stock prices, trading volumes, and interest rates are crucial for investment strategies and risk (Nelson et al., 2017).

Although Large Language Models (LLMs) have achieved remarkable success in various domains, they face significant challenges in time-series forecasting due to difficulties in extracting salient information from abundant temporal data. One mainstream solution involves contextual LLMs that incorporate sequences of historical data into the model's input to capture temporal dependencies (Jin et al., 2023; Yu et al., 2023). However, these models struggle with input length limitations and computational inefficiency, hindering their ability to model long-term dependencies effectively. Another approach is Retrieval-Augmented Generation (RAG), which allows LLMs to retrieve relevant information from external databases during generation (Aksitov et al., 2023). Yet, RAG faces challenges in time-series forecasting because its retrieval mechanisms are optimized for discrete textual data rather than continuous temporal data, making it difficult to align retrieved information with forecasting tasks and leading to suboptimal performance.

To address these challenges, we introduce **TimeRAG**, a novel retrieval-augmented approach specifically designed for time-series forecasting. A major difference between TimeRAG and previous RAG methods is that our method is the first to directly apply to continuous and complex temporal sequences. Yet a significant bottleneck arises due to the absence of explicit retrieval ground truths, unlike in textual data where relevant documents are clearly defined, making it challenging to train the retriever effectively. To overcome this, inspired by Zhang et al. (2023), we design a novel training target that leverages LLM feedback to guide the retrieval process. We utilize the generation probability of the LLMs for the correct tokens to determine positive and negative samples, which are then used for contrastive learning of the embedder. This approach aligns the retriever's prioritization with the LLM's assessments, bridging the gap between the information deemed important by the retriever and that recognized by the LLMs. Our methodology involves extracting sequences based on

the trained retriever, embedding them into the LLM's input context by formatting time-series data in JSON to reduce the comprehension gap, and using this enriched context along with the original query to perform forecasting, effectively combining salient historical patterns with current data to improve prediction accuracy.

We evaluate TimeRAG on the task of stock movement prediction using four benchmark datasets of high-trade-volume stocks in U.S. markets: ACL18 (2014-2015) (Xu & Cohen, 2018), BIGDATA22 (2019-2020) (Soun et al., 2022), and CIKM18 (2017-2018) (Wu et al., 2018). To assess prediction performance on more recent stock data, we construct a new dataset, Stock23, which includes stock prices from 2022 to 2023. This addition ensures that our evaluation reflects current market conditions, offering a more comprehensive benchmark for modern stock prediction tasks. Our experimental results demonstrate that TimeRAG significantly outperforms conventional context-learning LLMs and existing RAG methods. This superior performance is attributed to TimeRAG's ability to leverage LLM feedback to guide the retrieval process, effectively extracting and prioritizing historical sequences that enhance forecasting accuracy. By aligning the retrieval mechanism with the LLM's predictive objectives, TimeRAG captures both short-term fluctuations and long-term dependencies inherent in financial markets, overcoming challenges posed by data volume and noise.

Our contributions are summarized as follows:

1. We introduce TimeRAG, the first retrieval-augmented generation approach specifically tailored for time-series forecasting.

2. TimeRAG leverages LLM feedback to improve information retrieval, and employs an outcome-oriented approach to filter relevant data from extensive historical contents.

3. Experimental results demonstrate that TimeRAG outperforms previous contextual and RAG methods in accuracy for stock price movement prediction on four real-world datasets, showcasing its unique ability to identify and utilize the most impactful sequences for time-series forecasting.

## 2 PROBLEM DEFINITION AND GOALS

Time-series forecasting predicts future values or trends $G$ based on the given query sequence $q$ and retrieved sequences $c$, where all the sequences are collected sequentially over time at regular intervals. The goal is to model the retrieve model $R$ to efficiently retrieve useful information from a vast range of candidate sequences. In our finance example, the task is framed as a binary classification problem: predicting whether a stock's price will *rise* or *fall* on the next trading day. The model is given a query sequence $q$, which represents the stock's price over the previous $t$ days. Using this query, the model retrieves relevant price sequences as context and then predicts the stock's movement $M_{q,d}$ for the next trading day $d$. Table 1 defines the major symbols we use in this paper.

## 3 THE TIMERAG FRAMEWORK

Our method focuses on optimizing the retrieval stage to extract relevant content and seamlessly integrate it into LLMs. In the data construction phase (Section 3.1), we first preprocess time-series data and explore various features and prompts to maximize LLM performance. Then we use LLM feedback to identify the most effective data formats and content. During candidate selection (Section 3.2), we classify positive candidates based on high-performance feedback from the LLM, while the remaining data are treated as negative candidates. During training (Section 3.3), we employ knowledge distillation to teach the model how to distinguish useful time-series data (positive candidates) for a given query sequence, enabling more accurate and relevant retrieval. In this paper, we focus on stock movement prediction, but our approach can be applied to other time-series prediction tasks as well.

### 3.1 RAW DATA CONSTRUCTION

We utilize stock price data to perform stock movement prediction. First, we retrieve key stock price features from the Yahoo Finance API, including open price, high price, low price, adjusted close price, and volume. Next, we pre-process all stock price data into JSON format to improve the LLM's

Table 1: The definition of symbols.

| General Time-series Symbol | Stock Movement Prediction Symbol | Definitions |
| --- | --- | --- |
| $q$ | $q = \{q_{d-t}, ..., q_{d-1}\}$ | The query time-series data. In stock movement prediction, $q$ refers to the query stock price sequence of length $t$, containing stock price data from trading day $d - t$ to trading day $d - 1$. |
| $G(q)$ | $G(q, d) \in \{rise, fall\}$ | The final output $G$ given the query $q$. In stock movement prediction, $G(q, d)$ shows the generation $G$ of the query stock $q$ on the query trading day $d$, belonging to *rise* or *fall*. |
| $P(c) = LLM(O\|q, c)$ | $P(c) = LLM(M_d\|q, c)$ | The possibility $P$ of the LLM to generate an accurate output $O$ given the query sequence $q$ and a candidate sequence $c$. In stock movement prediction, $P(c)$ refers to generating the accurate movement $M$ on the query trading day $d$. |
| $\mathbb{C}_{\mathbb{P}} = \{c_i \mid i = 1, \ldots, k\}$ | | The set of top-k retrieved sequences as positive examples, where $P(c_i) \geq P(c_{i+1})$. |
| $\mathbb{C}_{\mathbb{N}} = \{c_i \mid i = k+1, \ldots, n\}$ | | The set of negative retrieved sequences, where $P(c_i) \geq P(c_{i+1})$. |
| $w_i$ | | The soft weight of the $i^{th}$ retrieved sequences, where $w_i = P(c_i), i = 1, ..., k$. |
| $R$ | | The retrieve model. |

ability to interpret time-series data (Fang et al., 2024; Singha et al., 2023; Yin et al., 2023). Finally, we explore different feature combinations and prompt designs to optimize the LLM's performance.

### 3.1.1 DATA PREPROCESSING

We start by preprocessing all stock price data into five-day sequences and creating a list for each feature accordingly. For each trading day, we use a one-day sliding window. Following Yoo et al. (2021); Soun et al. (2022), movements are classified as a rise if they are greater than 0.55% and as a fall if they are less than -0.5%, based on the adjusted closing prices. If the movement falls between 0.55% and -0.5% during continuous trading days, we classify it as a freeze. It's important to note that we don't predict the freeze movement in query sequences, we only use it to calculate the recent movement list. In this way, we filter out minor and statistically insignificant price movements, thereby focusing on more significant trends. An example of a processed sequence is as follows:

```
{
"data_index": 1000010,
"query_stock": "ABBV",
"query_date": "2014-06-13",
"movement": "rise",
"date_list": ["2014-06-06", "2014-06-09", "2014-06-10", "2014-06-11", "2014-06-12"],
"open_list": [55.32, 54.42, 53.14, 53.85, 54.22],
"high_list": [55.4, 54.88, 54.08, 54.7, 54.25],
"low_list": [54.89, 53.72, 52.29, 53.75, 53.46],
"close_list": [55.1, 53.84, 53.97, 54.23, 53.66],
"adj_close_list": [36.85, 36.0, 36.09, 36.27, 35.88],
"volume_list": [3449800, 6297500, 8414700, 5386800, 3941600],
"movement_list": ["freeze", "fall", "freeze", "freeze", "fall"]
}
```

### 3.1.2 PROMPT SELECTION

To design an effective prompt with the most useful features, we optimize three components: the task definition prompt, query sequence representation (selecting valuable features from the query se-

quence), candidate sequence representation (selecting valuable features from candidate sequences), and the order of these parts. To achieve this, we extract a toy dataset containing 5 rise sequences and 5 fall sequences, using all sequences from the same stock in the previous year as candidates. We then experiment with various prompts that include the task definition, each query sequence, and its corresponding candidate sequences, the detailed experiment is shown in Section 4.4. Using the probability $P(c)$ of generating the correct movement, we compute scores for each combination of task definition, query representation, and candidate representation. For each query sequence, we calculate the mean score of the top-3 $P(c)$ to assess prompt effectiveness. An example of a prompt trial is shown in Figure 1.

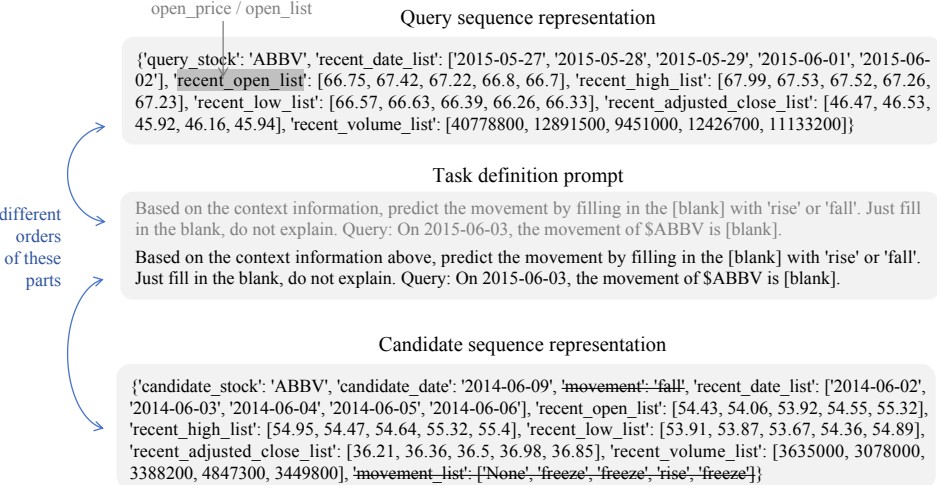

Figure 1: The prompt of TimeRAG.

**Task definition prompt** Utilizing the open-source LLaMA3-8b-instruct model, we first construct a fill-in-the-blank prompt designed to output only one token: 'rise' or 'fall'. This setup simplifies the calculation of the likelihood that the LLM generates the correct answer to the probability of the LLM producing the correct response at the first output index. Our selected task definition prompt is shown in Figure 1.

**Query sequence representation** We use the name of the stock and all recent price data to represent the query stock. We discuss how to list feature names to help the LLM better understand the referenced list. An example is highlighted in gray in Figure 1. In these trials, we name the list of open prices in the recent five days as 'open price', 'open list', or 'recent open list'. Once we definite feature names, we use the same name for candidate features.

**Candidate sequence representation** We discuss how different candidate features contribute to the prediction. We first provide all features and then provide the candidate sequence without each feature. After trials, we find the movement and recent movements of candidate sequences are noise for prediction. Therefore, we remove these two features, as is shown in Figure 1.

## 3.2 CANDIDATE SELECTION

In the last section, we confirmed the format and features we use in the query and candidate sequences. Then we need to select positive and negative candidate sequences for training. We consider all sequences from the past year of query stock as potential candidate sequences. Then we integrate the query sequence $q$ and each candidate sequence $c_i$ as the LLM input concurrently. The last step is to analyze the logits output by the LLM to calculate the probability of generating the correct response. The probability $P(c)$ indicates the probability of the LLM correctly predicting stock price movement $M_d$ for the queried trading day $d$.

We aim to train our retrieval model to retrieve sequences with a higher value of $P(c)$, thereby assisting the LLM in enhancing its prediction accuracy. To achieve this, we rearrange the candidate sequences in descending order according to $P(c)$, and select the top-1 sequences as a positive can-

didate and the last 15 sequences as negative candidates. The sets containing selected positive and negative sequences are denoted as $\mathbb{C}_\mathbb{P}$ and $\mathbb{C}_\mathbb{N}$. Moreover, the value of $P(c)$ serves as the teacher score, and we directly use it as the training reward for the corresponding candidate sequence $c$.

## 3.3 RETRIEVER TRAINING

Our retriever $R(q)$ is designed to intelligently distinguish between historically significant sequences $\mathbb{C}_\mathbb{P}$ and noisy sequences $\mathbb{C}_\mathbb{N}$, based on their support to the current query sequence $q$. Specifically, the process can be mathematically represented as:

$$R(q) = \arg\max_{s\in\mathbb{C}_\mathbb{P}\cup\mathbb{C}_\mathbb{N}} sup(q, s) \tag{1}$$

This formulation ensures the identification and extraction of sequences that maximize the measure $sup(q, s)$, where $\mathbb{C}_\mathbb{P}$ contains the top-k sequences with the highest scores, deemed most predictive of the future stock movement, and $\mathbb{C}_\mathbb{N}$ encompasses sequences with lower predictive utility. This approach not only facilitates more accurate and contextually rich predictions by focusing on the most informative historical sequences but also enhances the model's adaptability to evolving market conditions, thereby providing a robust framework for financial time-series analysis.

To train the retriever, we employ the pairs $(q, c_i)$ as soft labels. The samples within $\mathbb{C}_\mathbb{P}$ are treated as positive examples, while the candidates in $\mathbb{C}_\mathbb{N}$ are considered negative examples. To underscore the importance of the LLM outputting the correct price movement, we use the training reward as a soft weight, denoted as $w_i = P(c)$. It allows the model to weigh the training examples based on their likelihood of being correct. This nuanced approach ensures that the model pays more attention to sequences that not only are ranked higher but also have a higher probability of predicting the correct price movement, thereby fine-tuning its predictive capabilities.

To learn from soft rewards derived from the LLM, we conduct knowledge distillation. Particularly, we employ the KL-divergence to minimize the gap between the distributions of candidates computed using LLM's rewards and those predicted by the embedding model. In particular, for each query $q$ and its candidate list $\{\mathbb{C}_\mathbb{P}, \mathbb{C}_\mathbb{N}\}$, we derive the LLM's rewards towards the candidates, denoted as $\{P(c_i), i = 1, ..., n\}$. To make the LLM's rewards suitable for distillation, we transform each reward into a normalized weight: $w_i = softmax_R(P(c_i)/\alpha)$, where $\alpha$ represents the temperature. On top of these elements, the KL divergence is computed by the following equation:

$$\min. \sum_c -w_i \times \log\left(\frac{\exp\left(\langle \boldsymbol{e}_q, \boldsymbol{e}_{c_i}\rangle / \tau\right)}{\sum_{c'\in\mathbb{C}} \exp\left(\langle \boldsymbol{e}_q, \boldsymbol{e}_{c'}\rangle / \tau\right)}\right) \tag{2}$$

This loss function is designed to optimize the similarity between the query embedding and the embeddings of the top-ranked reference candidates, thereby enhancing the model's ability to predict stock price movements accurately.

## 4 EXPERIMENT

### 4.1 EXPERIMENTAL SETTINGS

**Datasets.** We evaluate TimeRAG on four benchmark datasets consisting of high-trade-volume stocks in US stock markets: 1) **ACL18** (Xu & Cohen, 2018) consists of 71 stocks along with their tweets and historical price data from 2014.06.02 to 2015.12.31; 2) **BIGDATA22** (Soun et al., 2022) consists of 47 stocks along with their tweets and historical price data from 2019.04.01 to 2020.12.31; 3) **CIKM18** (Wu et al., 2018) consists of 41 stocks along with their tweets and historical price data from 2017.01.03 to 2018.01. 23; 4) **Stock23** consists of 51 stocks along with their historical price data from 2022.01.03 to 2023.12.31. The detailed statistics are summarized in Table 2.

**Baselines** We evaluate whether the accuracy of the LLM's stock predictions is enhanced by incorporating example sequences selected through retrieval models, compared to approaches that use random sampling or no examples. We evaluate 4 retrieval methods in this setting: 1) **Instructor** (Su et al., 2023), a 1.5B instruction-finetuned text embedder. 2) **BGE** (BAAI general embedding) (Xiao et al., 2023), a 335M general embedder pre-trained from RetroMAE (Shitao Xiao, 2022). 3)

Table 2: Test dataset statistics.

| | stock amount | all sequences | | | | query sequences | | | |
|---|---|---|---|---|---|---|---|---|---|
| | | trading date | all | rise | fall | trading date | all | rise | fall |
| ACL18 | 33 | 2014.06.02-2015.12.31 | 7629 | 3840 | 3789 | 2015.06.03-2015.12.31 | 2690 | 1345 | 1345 |
| BIGDATA22 | 22 | 2019.04.01-2020.12.31 | 6534 | 3412 | 3122 | 2020.04.09-2020.12.31 | 2800 | 1400 | 1400 |
| CIKM18 | 19 | 2017.01.03-2018.01.23 | 2213 | 1228 | 985 | 2018.01.03-2018.01.23 | 80 | 40 | 40 |
| Stock23 | 51 | 2022.01.03-2023.12.31 | 19283 | 9627 | 9656 | 2023.01.03-2023.12.31 | 4128 | 2064 | 2064 |

**LLM-Embedder**, a 109M embedder fine-tuned with the feedback from LLMs. 4) **E5-mistral-7b-instruct** (Wang et al., 2023), a 7B embedder initialized from Mistral-7B-v0.1 (Jiang et al., 2023a) and fine-tuned on a mixture of multilingual datasets.

**Evaluation Metrics.** We employ Accuracy (ACC) and Matthews Correlation Coefficient (MCC) (Matthews, 1975) to assess the performance of TimeRAG and the baseline models on the stock movement prediction task. These metrics evaluate the performance of stock movement prediction based on the distribution of positive and negative samples. ACC and MCC are defined as:

$$ACC = \frac{TP + TN}{TP + TN + FP + FN} \tag{3}$$

$$MCC = \frac{(TP \times TN) - (FP \times FN)}{\sqrt{(TP + FP)(TP + FN)(TN + FP)(TN + FN)}} \tag{4}$$

where TP denotes true positives, TN denotes true negatives, FP denotes false positives, and FN denotes false negatives.

**Implementation Details** In our implementation, two key factors play a crucial role: the LLM foundation and the embedding model backbone. We choose LLaMA3-8b-instruct for feedback, as it is new, open-source, and powerful. For the embedding backbone, we use BGE, well-pretrained in general text embedding tasks, providing a strong foundation for TimeRAG.

## 4.2 MAIN RESULTS

Based on the results presented in Table 3, TimeRAG notably outperforms all evaluated approaches across the ACL18, BIGDATA22, CIKM18, and Stock23 datasets in terms of Matthews Correlation Coefficient (MCC). Specifically, TimeRAG achieves MCC scores of 0.140, 0.145, 0.197, and 0.219 respectively, which significantly surpass those of other methods. In contrast, the remaining baseline methods demonstrate much lower performance, with many yielding results close to random guessing, as indicated by their near-zero or negative MCC values. This strong performance of TimeRAG highlights its effectiveness in predicting stock movements accurately, underscoring the value of our method compared to traditional approaches and even advanced models like GPT-4 and LLaMA3-8b-instruct.

Table 3: Results of stock movement predictions using LLMs and retrieval models. The asterisk (*) indicates the LLM employed while using retrieval models.

| Methods | ACL18 | | BIGDATA22 | | CIKM18 | | Stock23 | |
|---|---|---|---|---|---|---|---|---|
| | ACC | MCC | ACC | MCC | ACC | MCC | ACC | MCC |
| LLaMA2-7B-chat | 0.500 | 0.010 | 0.499 | 0.000 | 0.500 | 0.056 | 0.500 | 0.000 |
| GPT-4 | 0.524 | 0.049 | 0.522 | 0.044 | 0.400 | -0.231 | 0.525 | 0.050 |
| FinMA-7B-full | 0.500 | 0.001 | 0.508 | 0.022 | **0.575** | **0.197** | 0.497 | -0.009 |
| LLaMA3-8b-instruct(*) | 0.522 | 0.048 | 0.497 | -0.006 | 0.475 | -0.070 | 0.527 | 0.067 |
| random retrieval* | 0.501 | 0.008 | 0.499 | -0.012 | 0.500 | 0.000 | 0.501 | 0.014 |
| Instructor* | 0.500 | 0.003 | 0.501 | 0.011 | 0.487 | -0.066 | 0.501 | 0.018 |
| BGE* | 0.501 | 0.005 | 0.502 | 0.015 | 0.475 | -0.095 | 0.501 | 0.015 |
| LLM-Embedder* | 0.508 | 0.025 | 0.501 | 0.003 | 0.512 | 0.052 | 0.511 | 0.055 |
| e5-mistral-7b-instruct* | 0.514 | 0.044 | 0.503 | 0.011 | 0.450 | -0.190 | 0.504 | 0.018 |
| TimeRAG* | **0.554** | **0.140** | **0.541** | **0.145** | 0.537 | **0.197** | **0.546** | **0.219** |

Despite significantly improved ACC in large sample datasets, our model also achieves positive MCC across all datasets, indicating that TimeRAG effectively retrieves valuable candidates to assist the LLM in analyzing stock sequences and predicting stock movements. Compared to GPT-4, our model's enhanced performance underscores the importance of these candidates. It indicates that using only the query sequence is insufficient to predict movements. Moreover, compared to our LLM foundation, LLaMA3, our results demonstrate the effectiveness of retriever training. Furthermore, our improvements over other retrieval methods highlight the benefits of task-oriented fine-tuning on stock data.

## 4.3 CASE STUDY

In this section, we investigate the ability of our retriever and baseline models to differentiate between various time-series data. We focus on the first example from the CIKM18 test dataset. As shown in Table 4, our retriever identifies positive candidates that are notably closer to the query sequence compared to the negative candidates, which are significantly more distant. This result highlights the superior retrieval performance of our method.

Table 4: A case of embeddings from different retrievers.

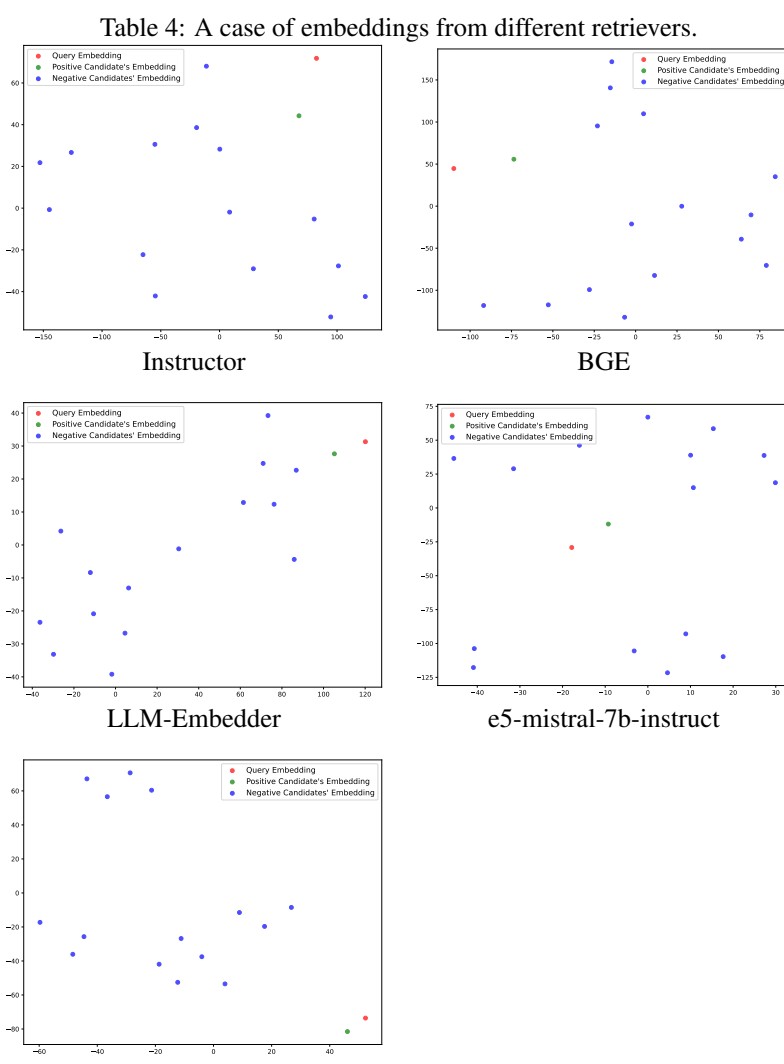

The proximity of the positive candidate to the query and the negative candidates are further away, implying that our model is effectively capturing the subtle patterns and dynamics in the time-series data, enabling more accurate retrieval. In contrast, baseline methods struggle to differentiate be-

tween similar but less relevant time-series sequences. This disparity illustrates the advantage of our retriever in isolating meaningful sequences for prediction tasks. Additionally, these findings reinforce the importance of a tailored retrieval strategy for time-series forecasting, where the subtle nuances in the data can significantly impact predictive accuracy.

## 4.4 ABLATION STUDY

In this section, we interpret how we conduct prompt selection shown in Section 3.1.2, by exploring the order of instruction, query sequence, and candidate sequence; exploring the name of features; and exploring which feature is important.

### 4.4.1 PROMPT SELECTION

Table 5 reveals that the sequence and content of prompts significantly impact the performance of stock movement prediction. The configuration marked as 6', which follows the 'qtc' order (query first, followed by task, then candidate) and includes the 'recent_xxx_list' feature without additional name and date details, scores the highest at 0.866. This indicates that specify temporal dynamics in the query significantly enhances prediction accuracy. It demonstrates that focusing on recent movement data in the query sequence and adhering to a structured prompt order optimizes the model's predictive capabilities. Thus, for higher prediction accuracy in stock movement, it is crucial to prioritize the incorporation of recent performance data and maintain a consistent structure in prompt arrangement.

Table 5: Scores for different prompts. *m list* refers to the recent movement list. Features labeled as *xxx_price* follow this naming format (e.g., *open_price*), while *xxx_list* indicates that features are named in the form of *xxx_list* (e.g., *open_list*).

| index | order | query sequence features | | | | candidate sequence features | | | | score |
| | | name | date | feature name | recent move-ment list | name | date | recent move-ment list | $d$ day's movement | |
| --- | --- | --- | --- | --- | --- | --- | --- | --- | --- | --- |
| 1 | tqc | w/o | w/o | xxx_price | w/o | w/o | w/o | w/o | w/o | 0.614 |
| 2 | tqc | w/o | w/o | xxx_list | w/o | w/o | w/o | w/o | w/o | 0.744 |
| 3 | tqc | w | w/o | recent_xxx_list | w/o | w | w | w/o | w | 0.629 |
| 4 | tqc | w | w | recent_xxx_list | w/o | w | w | w/o | w | 0.644 |
| 5 | cqt | w | w/o | recent_xxx_list | w/o | w | w | w/o | w | 0.783 |
| 6 | qtc | w | w/o | recent_xxx_list | w/o | w | w | w/o | w | 0.814 |
| 6' | qtc | w | w/o | recent_xxx_list | w/o | w | w | w/o | w/o | **0.866** |
| 7 | tqc | w/o | w/o | xxx_list | w | w/o | w/o | w | w/o | 0.756 |
| 8 | tqc | w | w/o | recent_xxx_list | w | w | w | w | w | 0.771 |

Another intriguing observation is that including the movement list and factoring in the movement of candidate data consistently results in lower scores. This suggests that LLMs predict stock movements by deeply analyzing the sequence data itself, rather than superficially following trends.

### 4.4.2 KEY CHARACTERISTICS FOR CANDIDATES

Table 6 presents an ablation study from prompt 6, analyzing the impact of removing various candidate features on the prediction score. The original score with all features included is 0.814. Removing the candidate data entirely results in a significant score drop to 0.530, indicating that candidate features are crucial for accurate predictions. Similarly, removing date and stock price data such as open, high, low, close, and volume also decreases the score, though to a lesser extent. The smallest score reductions occur when removing volume (0.083) and date (0.133), suggesting these features are less critical but still contribute positively to the model's performance.

Table 6: Score change when removing candidate features.

| prompt 6 | w/o candidate | w/o movement | w/o date | w/o open | w/o high | w/o low | w/o close | w/o volume |
| --- | --- | --- | --- | --- | --- | --- | --- | --- |
| 0.814 | 0.530 | 0.866 | 0.681 | 0.700 | 0.698 | 0.706 | 0.700 | 0.731 |
| | ↓ 0.284 | ↑ 0.052 | ↓ 0.133 | ↓ 0.114 | ↓ 0.116 | ↓ 0.108 | ↓ 0.114 | ↓ 0.083 |

Interestingly, removing movement information leads to a score increase to 0.866. This suggests that movement data might act as noise, distracting the model from more predictive patterns found in other sequence data. This finding implies that focusing on static features like price points and volume might enable a more robust analysis of stock movements, as these elements provide foundational data that the model can utilize more effectively than dynamic movement information.

## 5 RELATED WORK

### 5.1 TIME-SERIES FORECASTING WITH LLMs

To enhance the performance of LLMs in time-series forecasting, existing methods focus on the alignment of temporal and textual data, turning time-series into textual format, or encoding it and textual data into a unified vector space. For instance, Jin et al. (2023) reprogram time-series data into textual representations suitable for LLMs, enhancing prediction accuracy via declarative prompts. Similarly, Yu et al. (2023) and Liu et al. (2024) explore cross-modal alignment, with the former applying LLMs to financial forecasting using stock prices and news data, and the latter introducing a cross-modality framework to align time-series with text for improved predictive performance. Expanding on this, Pan et al. (2024) map time-series and text into a shared semantic space, further boosting LLM performance by strengthening data alignment. Despite advancements in time-series forecasting, many still require insights from extensive time-series data that cannot all be input into LLMs simultaneously. This limitation creates a need for retrieval-augmented methods, which our approach specifically addresses.

### 5.2 RETRIEVAL-AUGMENTED LLMs

To enhance LLM reasoning and prediction performance by retrieving relevant information from vast datasets, numerous retrieval methods have been proposed (Fan et al., 2024). Early approaches were based on keyword frequency, with many studies directly applying BM25 for passage-level retrieval in RAG (Chen et al., 2017; Jiang et al., 2023b; Ram et al., 2023; Xu et al., 2024; Zhong et al., 2022; Zhou et al., 2022). These passages were represented as bags of words and ranked using term frequency-inverse document frequency (TF-IDF) (Izacard & Grave, 2021). Later, methods based on semantic similarity emerged, encoding queries and passages into a unified vector space (Li & Qiu, 2023; Lu et al., 2023; Milios et al., 2023; Poesia et al., 2022; Rubin et al., 2022; Ye et al., 2023), intending to train embeddings to bring queries and factual passages as close as possible. However, these approaches are not well-suited for time-series retrieval, such as predicting stock price movements, where there are no fixed factual passages to retrieve. Moreover, due to the highly similar nature of time-series data, semantic similarity-based methods struggle to differentiate between them. Therefore, a specialized retrieval method for time-series data is required, which our model provides.

## 6 CONCLUSION

In this work, we present TimeRAG, a novel retrieval-augmented generation (RAG) approach designed specifically for financial time-series forecasting, with a focus on stock movement prediction. TimeRAG enhances the ability of large language models (LLMs) to interpret time-series data by integrating feedback mechanisms that address the lack of a clear retrieval ground truth. Our method bridges the gap between the information deemed important by the retriever and that recognized by the LLM, enabling a deeper understanding of market dynamics. We evaluate TimeRAG on four benchmark datasets of high-trade-volume stocks in the US markets—ACL18, BIGDATA22, CIKM18, and our newly constructed Stock23. Experimental results demonstrate that TimeRAG significantly outperforms conventional context-learning LLMs in prediction accuracy. This superiority is attributed to TimeRAG's unique ability to filter relevant time-series data from extensive and noisy historical datasets, employing an outcome-oriented retrieval approach that identifies sequences that most significantly enhance forecasting performance. Our findings underscore the potential of TimeRAG to advance time-series analysis in financial contexts, addressing the challenges faced by existing methods.

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
