# OpenReview forum: "TimeRAG: It's Time for Retrieval-Augmented Generation in Time-Series Forecasting"
_ICLR.cc/2025/Conference — ICLR 2025 Conference Withdrawn Submission_

### Official Review · Reviewer_CGmP · 2024-10-23

**Soundness:** 2
**Presentation:** 2
**Contribution:** 2
**Rating:** 3
**Confidence:** 4

**Summary:**

In this paper, the authors proposed a retrieval augmented generation for time series prediction, specifically on the stock movement prediction. The main challenge is the lack of ground truth data for retriever and the authors proposed to use outcome-oriented feedback from the LLM to guide the retriever. The retriever is optimized through contrastive learning with positive and negative samples defined by the LLM’s probability of generating correct trend. The authors showed experimental result that demonstrated the advantage of the proposed RAG framework compared to pretrained language models and other retrieval models.

**Strengths:**

Strength: this is the first work that attempts to address RAG for continuous, temporal signals.

The challenges for applying RAG on time series are well motivated and clearly explained.

**Weaknesses:**

Limitation: The scope of this paper is limited. This is not exactly time series forecasting, the title is somewhat misleading as the experiments are purely focused on stock price movement prediction. The baseline choices are irrelevant to time series forecasting.

Some choices of the setting is not well-justified: “select the top-1 sequences as a positive candidate and the last 15 sequences as negative candidates.” It’s not clear how this is decided.

“we don’t predict the freeze movement in query sequences, we only use it to calculate the recent movement list. In this way, we filter out minor and statistically insignificant price movements, thereby focusing on more significant trends.” the freeze situation also seems important as many people would like to know if the stock price will remain relatively stable.

The presentation of the paper can be improved such as adding a main figure for the overall pipeline. Additionally, in figure 1 the grey text are not explained. Is it a task definition prompt that was tried but discarded for use in the end. The only explanation is “An example is highlighted in gray in Figure 1. In these trials, we name the list of open prices in the recent five days as ’open price’, ’open list’, or ’recent open list’. Once we definite feature names, we use the same name for candidate features.” but this explanation is not concerned with the task definition prompt at all.

The outcome oriented model may lack generalizability to other applications.

**Questions:**

Questions:

- why is historical data of the same stock chosen as candidates instead of other stocks from the same & very recent periods. The intuition is that recent stock prices or other stocks from same period may have more influence on the future trend of the query stock. It seems counterintuitive that some distant past of the same stock would help prediction.
- In the candidate sequence representation section, it’s mentioned that “After trials, we find the movement and recent movements of candidate sequences are noise for prediction. Therefore, we remove these two features, as is shown in Figure 1.” However, it’s not clear what trials are done. How much data are used for these trials?
- I’m a little puzzled in the case study (figure 4), does retrieving the closest datapoint justify “model is effectively capturing the subtle patterns and dynamics in the time-series data, enabling more accurate retrieval.”? From my understanding, it could well be the case that the information retrieved could be complementary information not present in the input data and that would still effectively help the model and be considered good retrieval. This figure feels like the retriever acts like a KNN model (k=1).

---

> ### Author Response · Authors · 2024-11-18
> **Reply to Reviewer CGmP**
>
> Thank you very much for your valuable suggestions. We will revise our paper to clarify the motivation behind retrieving large amounts of time-series information to enhance LLM's performance on time-series forecasting tasks, emphasizing how traditional time-series methods are less effective in handling large amounts of time-series input. Additionally, we will add experimental details, figures and include experiments on other time-series forecasting tasks to further evaluate the validity and generalizability of our approach across diverse scenarios.

---

### Official Review · Reviewer_e9G8 · 2024-10-31

**Soundness:** 2
**Presentation:** 2
**Contribution:** 2
**Rating:** 3
**Confidence:** 4

**Summary:**

This work investigates a retrieval augmented generation-enhanced stock series prediction model using LLM feedback, addressing the absence of ground truth and aligning the priorities of the retriever and the LLM. Experimental implementations across four stock datasets including ACL18, BIGDATA22, CIKM18, and Stock23 have verified the effectiveness of proposed TimeRAG.

**Strengths:**

- S1. Good paper organization and it is easy to read.

- S2. The propsed TimeRAG can leverage the LLM feedback to train the retrieval, which employs the KL-divergence to minimize the gap between the distributions of candidates computed and those predicted by the embedding model, results in an outcome-oriented approach to filter relevant data from extensive historical contents.

- S3. The experimental results on four real-world datasets show the superiority of the proposed solution. And experiments also verify the motivation of RAG (retrieval-based generation).

**Weaknesses:**

- W1. The title of 'TIMERAG: IT’S TIME FOR RETRIEVAL-AUGMENTED GENERATION IN TIME-SERIES FORECASTING' may be not suitable, as the authors only investigate and implement experiments on only one task, i.e., stock series prediction. More experiments on various domains are required or the modification of the title is needed.

- W2. Lacking generalization capacity. The proposed TimeRAG requires re-training the model for each stock, results in the limited generalization capacity.

- W3. Short of literature review, evaluation baselines and issues on technical contributions. There are many baselines using LLM for time-series prediction, but have not been compared in this work. To name a few but not all, e.g., Time-LLM [1], LLMTime [2], GPT4TS [3]. More discussions on these abovementioned works and select some of them as baselines for comparisons are encouraged. Also, to my knowledge, there are other financial rag models such as AlphaFin [4] and FinGPT [5]; you should compare your model's performance with theirs. Therefore, the technical contributions of this work should be emphasized and distinguished.

- W4. This work only investigates the learnable scheme on retrieval side, but is there any designs for coupling the stock prediction with RAG from the LLM side or prompt side? Can learnable elements be inserted into the prompt side? More discussions or prospective solution should be presented.

[1] Jin M, Wang S, Ma L, et al. Time-LLM: Time Series Forecasting by Reprogramming Large Language Models[C]//The Twelfth International Conference on Learning Representations.

[2] Nate Gruver, Marc Finzi, Shikai Qiu, and Andrew G Wilson. Large language models are zero-shot time series forecasters. Advances in Neural Information Processing Systems, 36, 2024.

[3] Tian Zhou, Peisong Niu, Liang Sun, Rong Jin, et al. One fits all: Power general time series analysis by pretrained lm. Advances in neural information processing systems, 36:43322–43355, 2023a.

[4] Li X, Li Z, Shi C, et al. AlphaFin: Benchmarking Financial Analysis with Retrieval-Augmented Stock-Chain Framework[J]. arXiv preprint arXiv:2403.12582, 2024.

[5] Yang H, Liu X Y, Wang C D. Fingpt: Open-source financial large language models[J]. arXiv preprint arXiv:2306.06031, 2023.

**Questions:**

Please see my weaknesses.

**Details Of Ethics Concerns:**

N/A.

---

> ### Author Response · Authors · 2024-11-18
> **Reply to Reviewer e9G8**
>
> Thank you very much for your valuable suggestions. We will revise our paper to clarify the motivation behind retrieving large amounts of time-series information to enhance LLM's performance on time-series forecasting tasks, emphasizing how traditional time-series methods are less effective in handling large amounts of time-series input. Additionally, we will add experimental details and include experiments on other time-series forecasting tasks to further evaluate the validity and generalizability of our approach across diverse scenarios.

---

### Official Review · Reviewer_f7aV · 2024-10-31

**Soundness:** 2
**Presentation:** 1
**Contribution:** 2
**Rating:** 3
**Confidence:** 4

**Summary:**

In this paper, the author proposed a RAG approach to retrieve historical trades from other stocks to help LLM to better predict the trend of the target stock in the future. After preparing the positive candidates, a retriever model is trained. Experiments are conducted on real-world stock datasets and show the effectiveness of the proposed method.

**Strengths:**

1. The experiments are conducted on real-world stock datasets and the proposed method outperforms many baselines.

**Weaknesses:**

1. The presentation of the paper needs significant improvement. I feel it difficult to follow. Motivation is not clear and figures technical part are difficult to follow. See Q1-Q4.
2. The experimental settings are vague, which makes it difficult to reproduce the results. How the training split is conducted? Which are the hyperparameters and how did the authors tune them? See. Q5
3. Only LLM baselines are compared. The authors should include more non-LLM baselines, e.g. classic time series classification models. See. Q6-Q7.

**Questions:**

1. The motivation is not clear. With only numerical data, why can LLM do the job of stock trend prediction, if only a little text (e.g. open price, volume.) information is used?  Why not try some basic ML models for this task? What is the intuition behind using LLM? The authors are suggested to better motivate the paper with an example showing which kind information do LLM use in the RAG.
2.  Figure 1 is difficult to understand. There are two almost identical prompts in Task definition prompt. What are the relationships to the results shown above and below them. The authors are suggested to improve the presentation of the figure.
3. How do you derive P(c) in Section 3.2 specifically? Please give the detailed steps.
4. What do you mean by LLM's rewards? Please clarify.
5. The experimental setting is not clear. Which part of the datasets are used for training/validation/testing and hyperparameter settings? Please clarify.
6. Some good results are shown, but the intuition is not clear. What retrieved the content helped LLM to answer better? The case study is not helpful for understanding this. The authors should include more non-LLM baselines, e.g. classic time series classification models.
7. The term time series forecasting may not be accurate. It is more related to forecasting the exact value in the future. Time series trend forecasting may be more appropriate for the paper.

---

> ### Author Response · Authors · 2024-11-18
> **Reply to Reviewer f7aV**
>
> Thank you very much for your valuable suggestions. We will revise our paper to clarify the motivation behind retrieving large amounts of time-series information to enhance LLM's performance on time-series forecasting tasks, emphasizing how traditional time-series methods are less effective in handling large amounts of time-series input. Additionally, we will add experimental details and include experiments on other time-series forecasting tasks to further evaluate the validity and generalizability of our approach across diverse scenarios.

---

### Official Review · Reviewer_9wXT · 2024-11-04

**Soundness:** 3
**Presentation:** 3
**Contribution:** 2
**Rating:** 3
**Confidence:** 4

**Summary:**

This paper proposes a retrieval-augmented generation (RAG) framework specifically designed to improve time-series forecasting with large language models (LLMs). To address LLMs’ challenges with continuous temporal data, TimeRAG retrieves and embeds relevant historical data sequences in JSON format, making sequential dependencies more accessible to the model. This approach leverages LLM feedback to train the retriever, allowing it to prioritize historical sequences that align with predictive tasks such as stock movement forecasting. Evaluations across financial datasets demonstrate TimeRAG’s superiority over traditional LLM methods, suggesting its potential to enhance forecasting accuracy by focusing on task-aligned, contextually relevant data.

**Strengths:**

1. The "comprehension gap" TimeRAG addresses by structuring time-series data in JSON is tailored to LLM processing. This unique formatting approach is not emphasized in LLM-Embedder and is essential for making continuous data accessible to LLMs.
2. TimeRAG demonstrates its effectiveness specifically in stock movement prediction across multiple time-series datasets. This evaluation focuses on Matthews Correlation Coefficient (MCC) and Accuracy (ACC), which are metrics relevant for forecasting accuracy rather than general retrieval quality.

**Weaknesses:**

1. Problem Formulation Issue
    - Framing time-series forecasting as a binary classification task is unconventional and raises concerns regarding the model’s applicability. This approach simplifies the problem, reducing the predictive granularity necessary for effective forecasting.
    - Additionally, limiting predictions to only "rise" or "fall" cases ignores the frequent occurrence of "freeze" or neutral movements, a common scenario in real-world financial data. Excluding this category may significantly hinder the model’s practical utility and its ability to generalize across diverse market conditions.

2. Lack of Novelty Compared to LLM-Embedder
    - The methodology in TimeRAG closely mirrors the framework established in LLM-Embedder, lacking clear innovation. Key components, such as reward formulation based on LLM feedback, knowledge distillation using KL Divergence, and in-batch negative sampling (contrastive learning), are replicated with minimal adaptation. These similarities suggest limited original contribution.

3. Lack of Non-LLM Baselines in Time-Series Forecasting
    - The evaluation lacks comparisons with established time-series forecasting models outside the LLM domain. Forecasters like PatchTST and DLinear, widely regarded as effective baselines, are notably absent. Incorporating these models would provide a more comprehensive and rigorous performance benchmark, better situating TimeRAG’s effectiveness within the context of time-series forecasting research.

**Questions:**

1. Suggest switching the time-series forecasting task from classification to regression. This would allow the model to capture continuous trends in the data, which is generally more appropriate and beneficial for real-world forecasting applications.
2. In Table 4, the content appears to be graphical and might be better presented as a figure. Additionally, could you clarify whether t-SNE was used for dimensionality reduction in this visualization? This detail would help in interpreting the retrieval model’s embedding performance.

---

> ### Author Response · Authors · 2024-11-18
> **Reply to Reviewer 9wXT**
>
> Thank you very much for your valuable suggestions. We will revise our paper to clarify the motivation behind retrieving large amounts of stock-related information, emphasizing how traditional time-series methods are less effective in handling this aspect. Additionally, we will include experiments on other time-series forecasting tasks to further evaluate the validity and generalizability of our approach across diverse scenarios.

---

### Note · Authors · 2024-11-26

I have read and agree with the venue's withdrawal policy on behalf of myself and my co-authors.